# Clinical Relevance of Immunobiology in Umbilical Cord Blood Transplantation

**DOI:** 10.3390/jcm8111968

**Published:** 2019-11-14

**Authors:** Hyun Don Yun, Ankur Varma, Mohammad J. Hussain, Sunita Nathan, Claudio Brunstein

**Affiliations:** 1Division of Hematology, Oncology and Cellular Therapy, Department of Medicine, Rush University, Chicago, IL 60091, USA; hyun_don_yun@rush.edu (H.D.Y.); ankur_varma@rush.edu (A.V.); Mohammad_Junaid_Hussain@rush.edu (M.J.H.); sunita_nathan@rush.edu (S.N.); 2Division of Hematology, Oncology and Transplantation, Department of Medicine, University of Minnesota, Minneapolis, MN 60612, USA

**Keywords:** umbilical cord blood transplantation, NK cells, T cells, immune reconstitution, acute myeloid leukemia (AML), acute lymphoblastic leukemia (ALL), relapse, treatment-related mortality, leukemia free survival, overall survival

## Abstract

Umbilical cord blood transplantation (UCBT) has been an important donor source for allogeneic hematopoietic stem cell transplantation, especially for patients who lack suitable matched donors. UCBT provides unique practical advantages, such as lower risks of graft-versus-host-disease (GVHD), permissive HLA mismatch, and ease of procurement. However, there are clinical challenges in UCBT, including high infection rates and treatment-related mortality in selected patient groups. These clinical advantages and challenges are tightly linked with cell-type specific immune reconstitution (IR). Here, we will review IR, focusing on T and NK cells, and the impact of IR on clinical outcomes. Better understanding of the immune biology in UCBT will allow us to further advance this field with improved clinical practice.

## 1. Introduction

Since umbilical cord blood transplantation (UCBT) was first implemented for children and adults, it has been a valuable alternative donor source for allogeneic transplantation given its logistic advantages and comparable clinical outcomes to other types of hematopoietic stem cell transplantations (HCTs). Umbilical cord blood (UCB) grafts contain a unique cell composition in lymphocytes, and immune reconstitution (IR) of T and NK cells following UCBT appears to be somewhat different from other donor types. Here, we review immune cell composition in UCB, IR with focus on T and NK cells, and clinical relevance of IR in UCBT outcomes.

## 2. Lymphocyte Subsets in the UCB Graft 

The immune cell composition and properties of UCB units is different from peripheral blood or bone marrow. Functional and physiological relevance of the lymphocytes in UCB graft can be assessed by comparing to adult peripheral blood (PB) [1]. The absolute numbers of T, B, and NK cells per volume and the fraction of NK and B cells are higher in UCB than PB (Figure 1) [1]. The median T cells, NK cells, and B cells are 61%, 23%, and 16% in UCB, respectively, whereas 75%, 13%, and 12% in PB. In UCB, there are two lymphocyte populations (CD45^dim^ and CD45^bright^), whereas in PB, lymphocytes are all CD45^bright^ population [2]. The CD45^dim^ lymphocyte population contains higher fractions of B and NK cells than CD45^bright^ lymphocytes in UCB. The phenotypic and functional characteristics of T and NK cells in UCB grafts are further discussed below and summarized in Figure 2. 

### 2.1. CD3+ T cells 

As compared to PB, the UCB CD3^+^ compartment has a lower number of NK-T and TCRγδ^+^ T cells [1]. CD4^+^ and CD8 T cells are 72% and 28%, respectively, in UCB, whereas 65% and 35% in PB. The fractions of naïve (CD45 RA^+^/RO^−^) and memory T cells (CD45 RA^−^/RO^+^) are also different from PB (Figure 1); the median naïve T cells is 85% and memory T cells is 6% in UCB, whereas naïve T cells 39% and memory T cells 50% in PB. UCB contains a significantly greater percentage of CD45^+^/CD62L^+^ “recent thymic emigrants”, naïve CD4 and naïve CD8 T cell subsets with lower numbers of memory CD8^+^ T cells [1,3]. Cytotoxic T lymphocyte (CTL, CD8^+^/CD45RA^+^/CD27^−^) and suppressor T cell (CD8^+^/CD57^+^/CD28^−^) populations are absent, with a lower percentage of effector (CD8^+^CD11b^+^), activated (HLA-DR^+^), Th1-type (CCR-5^+^) T cells [1]. Cutaneous lymphoid antigen (CLA) is expressed on T cells preferentially homing to cutaneous inflammatory areas. Interestingly, CLA is not expressed on UCB T cells, potentially relevant to a lower incidence of GVHD in UCBT [1,4]. Upon phytohemagglutinin (PHA) and interleukin-2 (IL-2) stimulation, newborn CD4^+^CD45RA^+^ naïve T cells convert more rapidly into CD4^+^CD45RO^+^ memory T cells than the adult counterpart, suggesting a greater capability of UCB naïve T cells to transform into memory T cells [5]. However, IFNγ and TNF⍺, as well as IL-2 production of UCB lymphocytes, are markedly lower than adult PB lymphocytes [3], possibly due to the reduced expression of NFAT-dependent genes [6]. Notably, UCB T cells have powerful allogeneic antitumor activities with a higher tumor-infiltrating CD8/Treg ratio and rapid differentiation into memory/effector cells than adult PB T cells [7].

### 2.2. Tregs

Tregs are conventionally defined as a subset of CD4^+^CD25^+^ T cells that maintain self-tolerance and immune suppression. Forkhead box protein P3 (FoxP3) is the master transcription factor for development of Tregs [8,9]. Tregs are either developed in the thymus (nTregs) or transformed from CD4^+^CD25^−^ naïve T cells in the presence of transforming growth factor beta (TGFβ) in peripheral lymphoid tissues (iTregs) [8,10]. UCB possesses a distinct CD25^+^ population within the CD4^+^ T cell compartment [11]. The majority of UCB regulatory T cells (Tregs) are inexperienced with antigen stimulation, hence, cannot suppress “antigen-specific” alloreactive T cells [12,13]. However, after UCB Tregs are expanded and activated by TCR engagement and cytokine stimulation (e.g., stimulation with anti-CD3/CD28 mAb-coated beads and IL-2), their suppressive potency for allogeneic T cells become significantly greater than Tregs isolated from adult PB [11]. Furthermore, UCB Tregs possess a higher capacity for expansion [14], and Tregs (CD4^+^CD25^+^CTLA4^+^) are more readily inducible from CD4^+^CD25^−^ T cells in UCB than adult PB [13]. Notably, early phase clinical trials using ex vivo UCB-derived nTreg expansion have shown promising clinical efficacy to prevent acute GVHD [15,16].

### 2.3. NK Cells 

NK cells are known to be the main effector for graft-versus-leukemia (GVL) reactions early after HCT [17,18], and are enriched comprising up to 30% in the UCB graft (Figure 1) [1,19,20,21]. The CD56^bright^/CD56^dim^ ratio of UCB NK cells is similar or slightly higher than that of PB NK cells (Figure 1) [20,22,23]. Notably, the cytotoxicity of CD56^dim^ NK subset is poorer than that of CD56^bright^ NK cells in the UCB graft, whereas CD56^dim^ NK cells exert stronger cytotoxic effects in PB [22]. Cytotoxicity of CD56^bright^ subset of UCB NK cells is comparable to that of PB NK cells, whereas cytotoxicity of CD56^dim^ NK cells of UCB is greatly diminished compared to the counterpart of PB. A conjugate forming assay revealed that binding of CD56^dim^ NK cells of UCB to leukemic targets was significantly impaired secondary to a lower expression of adhesion molecules, including CD2, CD11a, CD18, and DNAM-1 [22]. In another report, L-selectin, ICAM-1 expression was significantly lower in NK cells of UCB than PB [20,23]. The expression of chemokine receptors in UCB NK cells are different from PB NK cells. CXCR1 expression in UCB CD56^dim^ NK cells is significantly reduced, whereas CXCR4 expression in both UCB CD56^dim^ and CD56^bright^ NK cells is increased [23]. Hence, UCB NK cells may be less responsive to inflammatory stimulation (involved in CXCR1 expression), but better capable of homing to the bone marrow (CXCR4), perhaps accounting for the effective GVL of UCBT [24]. Compared to PB NK cells, UCB NK cells less express maturation markers such as KIR, CD16, and CD57, whereas the expression of NKG2A/CD94, a phenotypic marker of NK cell immaturity, is higher in UCB NK cells [20,21,23]. In terms of NK cell activation receptors, the expression of NKG2C/CD94 and NKp46 (in CD56^bright^) is lower, but TLR-4, GITR, 2B4, and CD48 (in CD56^bright^) expression is higher in UCB NK cells than PB NK cells [2,23].

Immature CD56^−^/CD16^+^ NK cells are a distinct population identified in the UCB graft, but rarely found in healthy adults [18,25,26]. CD56^−^/CD16^+^ NK cells can differentiate into CD56^+^/CD16^+^ NK cells under IL-2 or IL-15 stimulation, acquiring enhanced cytotoxicity [26]. The expansion of UCB NK cells is less responsive to low-dose IL-2 stimulation (200 IU/mL) than PB NK cells. This is likely secondary to lower expression of CD25 (IL-2Rα) in CD56^dim^ NK cells in UCB [23]. While resting, UCB CD56^dim^ NK cells express lower levels of CD107a, IFNγ, granzyme B, perforin, and FAS-L, thereby exert a lower cytotoxicity than PB NK cells. UCB NK cells can acquire a potent cytotoxicity with an enhanced production of IFNγ and cytotoxic granules by high-dose IL-2 stimulation (1000 IU/mL) [23]. Furthermore, the response of UCB NK cells to IL-12 and IL-18 stimulation measured by IFNγ production and CD69 expression is higher than that of PB NK cells [27,28].

## 3. Immune Reconstitution in UCBT

Immune reconstitution after HCT may impact clinical outcomes such as incidence of transplant-related mortality and GVHD, the risks of infections and relapse, and, ultimately, survival. Numerous factors influence IR after HCT, including conditioning regimen (myeloablative vs. nonmyeloablative, use of antithymocyte globulin, and total body irradiation), immune suppression regimen, the graft source, cell compositions of the graft, and viral infections in HCT [29]. In this section, we will focus on T and NK cell reconstitution after UCBT, and compare with other donor types when possible. 

### 3.1. T cells 

T cell reconstitution is delayed after UCBT as compared to bone marrow transplantation (BMT) [30] and peripheral blood stem cell transplantation (PBSCT) [31]. T cell reconstitution after HCT occurs in two distinct pathways: (1) Peripheral expansion of mature T cells (thymus-independent pathway), and (2) thymopoiesis from donor hematopoietic progenitors (thymus-dependent pathway) [32,33]. Early after HCT, T cell reconstitution takes place through peripheral expansion by T cells transferred from the graft or recipient T cells which have survived conditioning therapy (*thymus-independent pathway*). The mature T cells compete for homeostatic cytokines, such as IL-7 or IL-15, and self-MHC molecules presented by antigen-presenting cells (APCs) [34]. In the lymphopenic condition, IL-7 and IL-15 are constantly produced by immune and non-immune cells, but little is consumed. Hence, there are high plasma levels of these cytokines early after HCT [35]. The mature T cells transferred from the graft have greater access to IL-7 or IL-15 and self-MHC on APCs, which, in the context of lymphopenia, promotes expansion with limited competition [34,36]. Naturally, the T cells undergo rapid expansion with multiple cell divisions, leading to accelerated telomere shortening in the first year post-HCT [37]. In contrast to memory T cells, naïve T cells require TCR engagement with MHC molecules presented by APC, in addition to cytokine stimulation, for survival and expansion [34,38]. As a result, peripheral expansion of memory T cells is greater than naïve T cells after HCT. Another unique aspect of early immune reconstitution is the inverted CD4/CD8 ratio, secondary to better peripheral expansion of memory CD8 T cells [29]. In certain donor/recipient pairs, seropositivity to viral pathogens like CMV can polarize T cell expansion towards the viral antigens, narrowing the T cell repertoire that predominantly proliferates towards antigen-specific memory T cells, limiting polyclonal expansion [39]. In UCBT, early T cell reconstitution is primarily dependent on the peripheral expansion and may have limited T cell repertoire due to delayed thymopoiesis (especially in adults), as compared to other donor types [40]. Furthermore, CD4^+^ and CD8^+^ T cells at day +100 after UCBT have reduced capability to produce IFNγ upon superantigen and CMV stimulation, possibly indicating impaired T cell functions early after UCBT [41]. In a pediatric study using myeloablative conditioning (MAC) with antithymocyte globulin (ATG), median time to T cell recovery (CD3^+^ T cells >0.5 × 10^9^/L) was 6.3 months in UCBT group vs. 3.2 months for unrelated BM group (*p* = 0.008) [30]. This was apparently driven by CD8^+^ T cell reconstitution (>0.25 × 10^9^/L) that took a median of 7.7 months after UCB vs. 2.8 months after unrelated BM. However, recovery of the CD4^+^ T cell (CD4^+^ T cells >0.5 × 10^9^/L) numbers was faster after UCBT, with median time for CD4^+^ T cell recovery 9.3 months vs. 12 months in unrelated BMT (*p* = 0.003). Hence, the inverted CD4/CD8 ratio is not observed early after UCBT because of the delayed CD8^+^ T cell recovery after UCBT [42]. 

The reconstitution of T cell repertoire diversity from donor-derived naïve T cells occurs in the thymus following peripheral expansion of mature T cells post HCT (*thymus-dependent pathway*). This process, termed thymopoiesis, requires a longer period of time in which donor-derived lymphoid progenitors enter the thymus and undergo maturation processes (positive and negative selection) [43]. Thymopoiesis occurs weeks after HCT and can last up to 6 years [44]. After the sequential positive and negative selections, only small fractions of T cells can survive and exit the thymus, so-called recent thymus emigrant (RTE) [45]. This de novo process can be measured by T cell receptor (TCR) excision circle (TREC), naïve T cell counts, and T cell repertoire diversity [44,46,47], and is critical for broad and self-tolerant T cell immunity [32,46]. After UCBT, TREC levels correlate with CD3^+^CD4^+^45RO^−^ naïve T cell counts (*r* = 0.83, *p* = 0.0001), and TCR repertoire diversity (*r* = 0.83, *p* = 0.0001) [48]. Long-term T cell reconstitution (CD3 >1.5 × 10^9^/L) is similar between UCBT and unrelated BMT (9.3 vs. 10 months) in the pediatric population [30]. In an age- and GVHD-matched comparison of children and young adults between UCB [median age 12.6 years (3–34.6)] and matched sibling recipients, TREC and CD4^+^CD45RO^−^ naïve T cells were significantly higher, whereas CD8^+^ activated and memory T cells were lower at 2 years in UCB as compared to matched sibling donor group, indicating efficient thymopoiesis in UCBT [48]. 

### 3.2. NK Cells 

Natural killer cells are the first lymphocytes reconstituting after HCT. NK cell immunity plays a critical role in GVL, especially early after UCBT, because of the low absolute counts and functional immaturity of T cells transferred with the UCB graft. The time to NK cell reconstitution (>0.1 × 10^9^/L) was similar between UCBT (1 month) and unrelated BMT (1.4 months), when both groups received ATG as part of the conditioning regimen [30]. Notably, after UCB with no ATG in the conditioning regimen, NK cell count reconstitution at 1 month after UCBT was similar to healthy controls [49,50]. Moreover, a better NK cell reconstitution with higher NK cell counts was observed over a 24-month period in UCBT than PBSCT [31,51].

NK cell reconstitution 1–3 months after UCBT is polarized to CD56^bright^ NK cells (approximately 40% of the total NK cells), as compared to healthy donor controls [49,50]. Three months after UCBT, NK cells express high levels of NKG2A and CD62L and low levels of CD16, CD8, and CD57 [49]. Even in CD56^dim^ NK cells, the expression of CD94/NKG2A, an inhibitory receptor recognizing HLA-E antigen, is higher early after UCBT, but gradually returns to levels similar to that of healthy controls’ by 1 year after UCBT [50]. The expression of KIR2DL2/3 and KIR3DL1 of NK cells is significantly lower in the UCB graft, but becomes comparable within 3 months after UCBT to healthy donors, indicating acquisition of NK cell education [49,50]. However, KIR2DL1 levels of CD56^dim^ NK cells are persistently lower than that of healthy controls during the first 6 months after UCBT, consistent with the sequential acquisition of KIR commonly observed in other types of HCT [49,50,52,53]. Interestingly, NKp30, NKp46 (natural cytotoxicity receptors involving NK cell activation), and CD69 (an activation marker) of CD56^dim^ NK cells are transiently higher for the first couple of months after UCBT than healthy controls [50], potentially providing advantages in GVL reactions. The HLA-DR expression of NK cells is significantly higher during the first year of UCBT than that of healthy controls and UCB grafts [49]. DNAM-1 (an activating NK cell receptor) expression of CD56^dim^ NK cells is significantly lower in the UCB graft, but gradually increases, and becomes similar to the level of healthy control NK cells within a year after UCBT [50].

NK cells acquire unique functional characteristics after UCBT, as evidenced by high IFNγ production in the first 1–3 months [49,50]. Direct cytotoxicity of NK cells during the first 6 months post-UCBT against K562 targets and HLA mismatched primary acute myeloid leukemia (AML) samples is robust, and similar to that of healthy controls [50]. However, antibody-dependent cellular cytotoxicity (ADCC) of NK cells within 3 months after UCBT is significantly impaired [50], consistent with low expression of CD16 early after UCBT [49].

## 4. Clinical Factors Associated with Immune Reconstitution in UCBT

As summarized in Figure 3, multiple clinical factors potentially influence the reconstitution of T and NK cells after UCBT. Selected factors are reviewed below. 

### 4.1. Viral Infections 

High incidence of HHV-6 (up to 70–80%) has been associated with delayed engraftment after UCBT in multiple reports [54,55]. HHV-6 infection can also interfere with T cell reconstitution, both in a thymus-dependent and independent pathways, resulting in dysfunctional T cell population after HCT [56]. A recent retrospective study with time-dependent analysis revealed that high HHV6 viral load (>10^5^ copies/mL) was associated with impairment of both CD4^+^ and CD8^+^ T cell reconstitution after HCT, including UCBT [57]. Interestingly, HHV6 infection negatively affected reconstitution of naïve, but not effector memory CD4^+^ T cells. In addition, reactivation of HHV-6 early after UCBT was associated with T cells expressing CD57, NKG2A, and KIR2DL2/3, surface markers of T cell senescence, and hypofunction, and it was associated in inferior clinical outcomes [58]. Other viral infections may result in limited and skewed TCR diversity towards a specific viral antigen, as observed in individuals with Epstein-Barr virus (EBV) and cytomegalovirus (CMV) infections than those without these infections [59].

### 4.2. GVHD 

Glucocorticoid (GC) is the cornerstone of treatment for acute and chronic GVHD. GC is known to induce in vivo Treg expansion [60,61], and inhibit the *JAK-STAT* signaling pathway induced by IL-2, IL-4, IL-7, and IL-15 in T cells [62]. A murine HCT model suggests CD8^+^ T cells as the main target of GC [63]. Taken together, CG used for treatment of GVHD profoundly affects in the T cell function. GC also prevents upregulation of MHC class II and costimulatory molecules on dendritic cells [64,65], hence, compromises T cell responses against foreign or allo-antigens. High-dose GC can also induce T cell apoptosis [65]. Furthermore, GVHD can directly damage the thymus by inducing apoptosis of thymocytes (thymic GVHD) [66,67]. Among allo-HCT recipients, absolute counts of naïve T cells were significantly lower at 12 months, with narrower and more skewed TCR repertoires in patients with aGVHD than without aGVHD [47,48]. Furthermore, both sjTREC and βTREC were significantly lower in the aGVHD group, but sj/βTREC ratio was comparable between groups with and without aGVHD, indicating that aGVHD primarily impairs early-stage thymopoiesis [47]. In addition, patients with cGVHD developed markedly lower TREC [44,48]. Interestingly, steady increases in TREC levels in both CD4^+^ and CD8^+^ T cells were observed in UCBT patients receiving immunosuppression without active GVHD, suggesting that GVHD prophylaxis with immunosuppression alone does not necessarily cease thymopoiesis [44].

### 4.3. Conditioning Regimen 

ATG has been frequently used in the conditioning regimen in UCBT. ATG delays T cell reconstitution by depleting naïve and memory T cells transferred with the graft required for early peripheral expansion of mature T cells [68]. In contrast, ATG exposure is associated with strong recovery of B and NK cells 30 days after transplant, enabling B and NK cells to compensate T cell defects in UCBT [41,43]. In addition, the timing, dose of ATG administration, and serum level at the time of the allograft infusion may influence incidence and grade of GVHD and T cell subset reconstitution [68,69]. In the absence of exposure to ATG, better T cell reconstitution after UCBT is observed, which may contribute, at least in part, to better leukemia control and lower all-cause mortality [68,70,71].

Reduced intensity regimen (RIC, Fludarabine 30 mg/m^2^ for 5 days, Cyclophosphamide 50 mg/kg, and TBI 200cGy) without ATG in adults employed at the University of Minnesota was associated with comparable lymphoid reconstitution at day 180 post-HCT and significantly lower chronic GVHD at 1 year post-HCT in UCBT compared to matched sibling donor HCT. In a pediatric UCBT study where all patients received ATG in the conditioning [72], NK cell counts after transplant were higher in the MAC group. Multivariate analysis revealed that the MAC group had a higher risk of developing acute GVHD (HR 6.1, *p* = 0.002), increased treatment-related mortality (TRM) (OR 26.8, *p* = 0.008), and overall mortality (HR = 4.1, *p* = 0.0001). In another adult study [73], CD3^+^ T cell recovery was observed at 6–12 months after UCBT with higher number of CD45RA^+^ T cells, more diverse T cell repertoire in patients treated with nonmyeloablative regimen (NMA) (fludarabine 30 mg/m^2^, cyclophosphamide 500 mg/m^2^, and ATG 30 mg/kg) when compared to a historical MA group [40]. Furthermore, TREC was detected at 12 months in NMA group, whereas at 18–24 months in the MA group [40,73]. Taken together, RIC may provide better T cell reconstitution, whereas MAC may favor NK cell reconstitution. 

### 4.4. Age 

Age is one of the most significant host factors to influence T cell reconstitution and thymopoiesis following UCBT. In adult patients with TBI- and ATG-based conditioning regimen, the number of CD8^+^ T cells reached normal ranges a year after UCBT, but total T cell counts remained below normal for 2 years from UCBT, and memory T cells remained 70% of the total T cell population until 12 months post-UCBT [40,41]. These data suggest that T cell reconstitution and thymopoiesis are delayed in adult UCBT. In contrast, the long-term immune recovery, including after UCBT, was similar to those who underwent adult unrelated donor [30,74] and haploientical donor HCT [75] in children, suggesting faster thymus-dependent T cell reconstitution. In a pediatric population, TREC numbers recovered to the pre-UCBT levels by 6 months [75], and reached within normal limits by 1 year [40,45], whereas in adults, sjTREC recovery took a median of 3 years after UCBT [40,41]. Furthermore, diverse T cell repertoires were observed at 1–2 years post-UCBT in children, whereas it took 3–4 years in adult patients [40]. Hence, in part, driven by reduced thymopoiesis, the delay of T cell numbers and repertoire reconstitution is a challenge in adults undergoing UCBT.

### 4.5. Cell Dose: CD34^+^ Progenitor Counts in Grafts and Single vs. Double Unit(s) of UCBT 

CD34^+^ count (>10^7^/kg vs. <10^7^/kg) in the allograft correlates with the TREC levels after HCT in children, suggesting that CD34^+^ cell dose plays a role in thymopoiesis [76]. However, despite a higher combined CD34 cell dose in children receiving double (dUCBT) UCB grafts, reconstitution of lymphocyte subsets was similar to that of single UCBT (sUCBT) in children (and young adults) at 1–2 years after UCBT [77,78]. While data directly comparing single with double UCBT in adults are not available, in adult dUCBT without ATG, the TCR diversity measured by TCR deep sequencing at 6 months after HCT approached that observed in a healthy control group [59]. Others reported that, despite administration of ATG, adult patients undergoing dUCBT had a steep rise in TREC numbers between 6 and 12 months after transplant [79], while the recovery of T numbers between 6 and 12 months approached that of recipients of sibling and unrelated donors grafts in the absence of ATG [80,81]. In summary, dUCBT seems to provide better T cell reconstitution in adults. At least in part, this may be explained by the “threshold effect” of CD34^+^ cell doses in thymopoiesis [76] (i.e., a single UCB graft may contain sufficient CD34^+^ progenitors to reach the “threshold” of thymopoiesis for pediatric but not for adult recipients). This effect, however, has to be considered in the context of conditioning regimen intensity and administration of ATG. 

## 5. Clinical Impacts of Immune Reconstitution on Outcomes of UCBT

Reconstitution of lymphocyte subsets after UCBT is influenced by multiple clinical factors, as discussed above. Conversely, immunological and genetic characteristics of T and NK cells are critical determinants of clinical outcomes in HCT. Figure 4 summarizes the potential impact of lymphocyte reconstitution on UCBT outcomes.

### 5.1. Infections 

Infection is a major cause of death in UCBT. Szabolcs et al. reported that 58% of deaths within 6 months in UCBT were due to infection in children [82]. A better T cell reconstitution at day 50 post-UCBT, including higher absolute CD4^+^ T cell counts, was observed in the group without subsequent opportunistic infections (OI) than with OI at day 100 post-UCBT. This finding indicates that T cells play a critical role to prevent opportunistic infection. Additionally, both CD34^+^ and CD3^+^ cell doses were associated with lower death from OI at 6 months. In a retrospective study where UCBT recipients constituted more than a half of the entire cohort, delayed CD4^+^ T cell reconstitution was predictive of adenovirus, EBV, and HHV6 infections [83]. Moreover, CD4^+^ T cell reconstitution (≥50 × 10^6^/L) within 100 days was associated with a shorter duration of adenovirus infection. CMV infection is a major life-threatening complication in HCT. In a large cohort study (*n* = 332) at the University of Minnesota, 51% of recipients with hematological malignancies undergoing UCBT developed CMV reactivation [84]. CMV-specific CD8^+^ T cells transferred from the UCB graft alone could not eradicate CMV viremia, but clearance of CMV viremia occurred later and seemed to depend on CD4^+^CD45RA^+^ T cells by thymopoiesis [79]. However, the Seattle group identified intact CMV-specific T cell priming early after UCBT (at day +42), suggesting that failure to control CMV reactivation is likely due to insufficient numbers of these T cells in vivo [85]. Nevertheless, successful T cell reconstitution by thymopoiesis is required for an optimal control of CMV reactivation. Additionally, NK cells may play an important role in CMV control. A recent study demonstrated that low NKG2C copy number of NK cells in the UCB graft was independently associated with increased risk of developing CMV reactivation (HR = 2.72, *p* < 0.0001) [86].

### 5.2. Major Clinical Outcomes: Relapse, Mortality, and Survival

Poor T cell reconstitution is associated with increased risks of relapse in HCT. Clave et al. reported that relapsed pediatric patients with hematological malignancy had lower β-TREC levels at 6 months before and after HCT (including UCBT), suggesting an association between impairment of early intra-thymic T cell development and increased relapse risks [75]. Regardless of graft types, both low CD4^+^ and naïve T cell reconstitution are significantly associated with increased risks of treatment-related mortality at day 100 post-HCT [87]. Successful CD4^+^ T cell reconstitution at day 100 after UCBT by less exposure to ATG is associated with lower non-relapse mortality (NRM), lower relapse-related mortality (particularly for AML), better event-free survival, and better overall survival (OS) [68]. Furthermore, early CD4^+^ T cell reconstitution (defined by CD4^+^ T cell >50 × 10^6^/L within 100 days after UCBT) resulted in better leukemia-free survival (LFS) (HR = 0.24, *p* = 0.003), improved OS (HR = 0.16, *p* = 0.0014) with lower NRM (HR = 0.20, *p* = 0.0072). Again, lower RI (HR = 0.31, *p* = 0.041) in association with improved T cell reconstitution was observed in AML [88]. Successful CD8^+^ T cell reconstitution and high TREC levels, as well as CD4^+^ T cell reconstitution, are associated with improved OS [79]. Moreover, CMV-specific T cell response and high NK cell counts are independently associated with better progression-free survival (PFS). As T cells in the UCB graft are unlikely to have encountered antigens of herpes viruses, herpes antigen-specific T cell response represents T cell reconstitution occurring in vivo. Hence, Parkman et al. measured herpes antigen-specific proliferative T cell responses as measurement of successful T cell IR [89]. In pediatric patients with acute leukemias undergoing ATG-based MA sUCBT, the earliest herpes-antigen specific T cell response was observed within the first month. Notably, negative antigen-specific T cell response was independently associated with higher leukemia relapse (HR = 3.7, *p* = 0.003) and lower relapse-free survival (HR = 3.6, *p* = 0.0002), indicating that successful T cell reconstitution plays a critical role in relapse prevention. 

Transplant outcomes in UCBT are also tightly associated with NK cell IR. In RIC UCBT for AML, the low CD16 and high HLA-DR expression on NK cells are significantly associated with increased risks of TRM [49]. In addition, KIR-HLA typing is associated with overall survival (OS) [49]. HLA C2 homozygous recipients have much poorer event-free survival (EFS) (HR = 6.19, *p* = 0.002), OS (HR = 6.12, *p* = 0.001), and higher TRM (HR = 9.44, *p* = 0.026) than HLA C1/x recipients. Furthermore, poor direct cytotoxicity of NK cells against K562 measured by CD107a expression was significantly associated with poor overall survival as well.

## 6. Closing Remarks

UCBT has been a valuable alternative donor for transplantation for the past several decades. It has clinical advantages, including readily available grafts, relatively lower incidence of GVHD, and lower disease relapse [4,24,77,90]. While there have been some challenges in UCBT, advances have been made by modifying conditioning regimens [87,91], double UCB grafts [77]. More sophisticated utilization of ATG has substantially improved clinical outcomes [68,69,70]. Emergence of more robust ex vivo expansion techniques have enabled to meet the adequate cell doses for larger patients [92,93]. In addition, a recent report described the powerful GVL effect of UCB in those with acute leukemia and minimal residual disease [24]. Better understanding of the immune biology in UCBT will lead to improved graft engineering in the future.

## Figures and Tables

**Figure 1 jcm-08-01968-f001:**
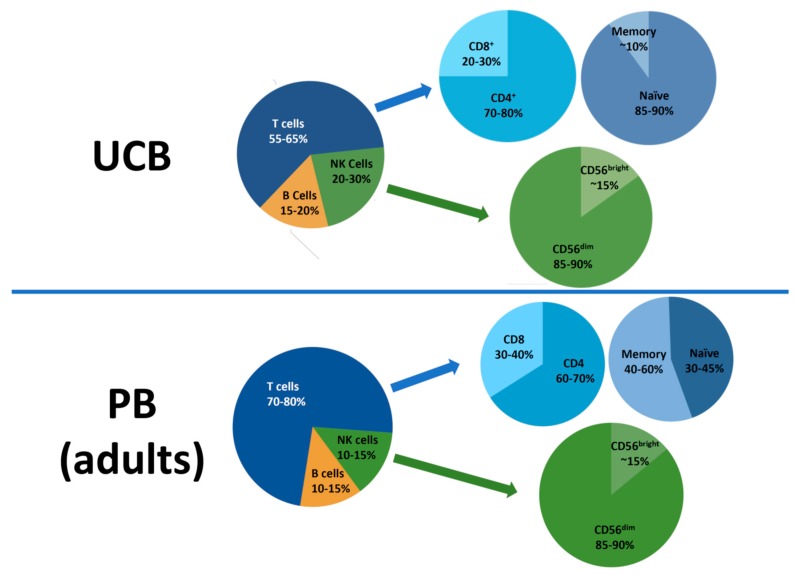
Cell composition in the umbilical cord blood (UCB) graft. The majority of T cells are CD4^+^ and naïve T cells in the UCB graft. The ratio of CD56^dim^/CD56^bight^ NK cells is similar to that of PB. However, the cytotoxicity of CD56^dim^ NK cells in the UCB graft is poor compared to that of PB NK cells. UCB, umbilical cord blood; PB, peripheral blood; CD, cluster of differentiation; NK, natural killer.

**Figure 2 jcm-08-01968-f002:**
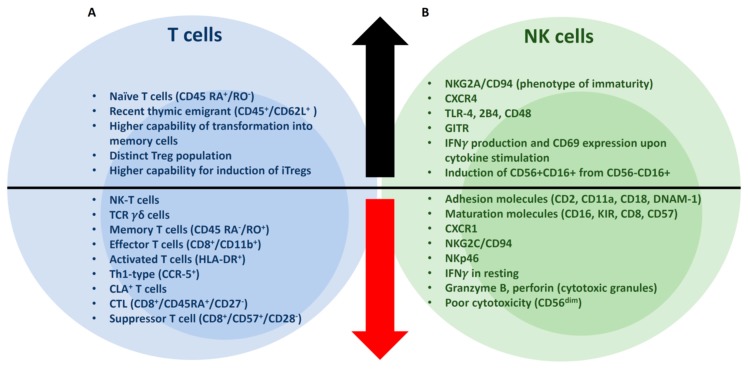
The immunophenotypic and functional characteristics of T and NK cells in the UCB graft. (**A**) Naïve T cells, the major subset of UCB T cells have a better plasticity to transform into memory T cells and iTreg cells than PB T cells. The UCB graft contains a distinct Treg population. The fractions of memory, effector, activated T cells are low in UCB graft. (**B**) NK cells of the UCB graft are characterized by the phenotype of immaturity. The cytotoxicity of resting UCB NK cells is markedly low. However, the UCB NK cells acquire potent cytotoxicity with phenotypic maturation upon cytokine stimulation.

**Figure 3 jcm-08-01968-f003:**
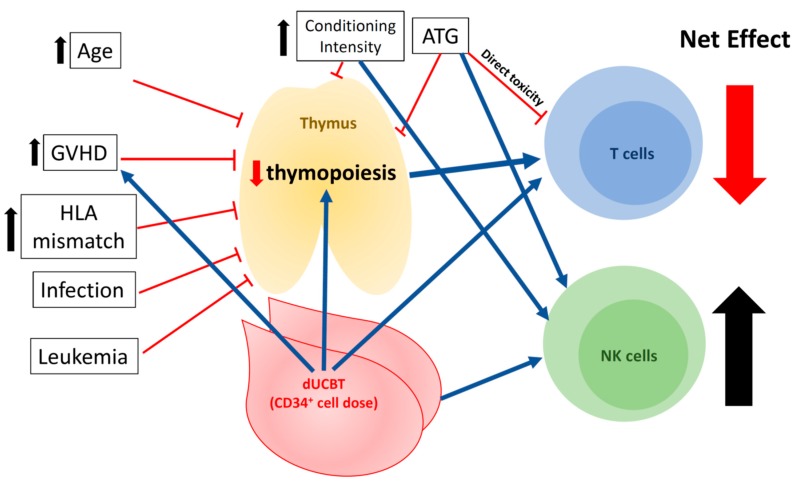
Impacts of clinical factors on immune reconstitution of T and NK cells in UCBT. Advanced age, acute and chronic GVHD, HLA mismatch, infection, myeloablative conditioning, and ATG result in impaired thymopoiesis. Furthermore, relatively low T cells in the UCB graft cause delayed T cell recovery. Double UCBT enhances early T cell IR and thymopoiesis (in adults). Contrarily, NK cell IR in UCBT is comparable or better, compared to other types of HCT, given relatively high fractions of NK cells in the UCB graft. ATG and myeloablative conditioning are known to further prompt NK cell IR. GVHD, graft-versus-host disease; UCB, umbilical cord blood; UCBT, umbilical cord blood transplantation; NK, natural killer; IR, immune reconstitution; HCT, hematopoietic cell transplantation.

**Figure 4 jcm-08-01968-f004:**
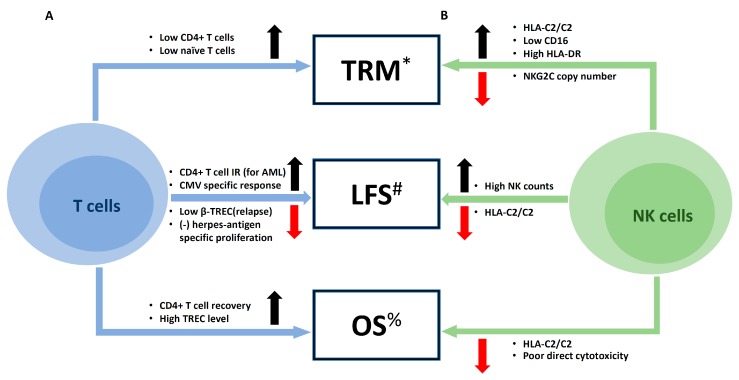
Impacts of T and NK cell IR on major clinical outcomes in UCBT. (**A**) Impacts of T cell IR on clinical outcomes. CD4^+^, naïve T cell IR, thymopoiesis measured by TREC, viral antigen-specific T cell immunity significantly impact clinical outcomes. (**B**) NK cell effects on clinical outcomes in UCBT. Immunogenetic factors such as recipients’ HLA typing play a critical role in clinical outcomes in UCBT by KIR–HLA interactions. Furthermore, the NK cell function (e.g., cytotoxicity against K562) and the phenotype (CD16, HLA-DR expression) can affect clinical outcomes. There are conflicting data in the literature on effects of KIR-L mismatch in clinical outcomes. * treatment-related mortality; ^#^ leukemia-free survival; ^%^ overall survival.

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
