# Peer review of "Clinical Relevance of Immunobiology in Umbilical Cord Blood Transplantation"

_jcm, 2019, doi:10.3390/jcm8111968_

Round 1

Reviewer 1 Report

This is an interesting and important review focused on the clinical relevance of immunobiology in UCBT. The review points out key differences in the immune cells in UCB and how the unique immune reconstitution (IR) following UCBT influences outcomes transplantation. The manuscript is nicely organized from discussing the composition of UCB, IR and outcomes following UCBT with specific focus on clinical factors that influence IR.

It is essential to highlight differences in the composition of the UCB graft compared to other transplant sources, such as PBSC (and specifically G-CSF-mobilized PBSC) and BM grafts, however the comparison is made to adult PB (not specified or mentioned re: G-CSF stimulation), so the comparison is not valid and should be revised for each lymphocyte subset appearing in section 2. In addition, as children and adults have different lymphocyte subsets, these should also be mentioned in the sections, to clarify whether the reference is to children vs. adult PBSCs vs. BM.

Figure 1 highlights the comparison, in which case it should include the composition of UCB, PBSC and BM. Re: the T cell subsets - naive vs. memory - it is not clear whether the pie charts refer to CD4 cells only or CD8 cells - it would be best to include both and also include other established T cells subsets, including stem cell memory, effector memory and terminal effector T cells.

A table listing the immunophenotypes of the different cell types would be helpful to include in the review.

The review mentions ratios of NK cells, and later, CD4 and CD8 T cells, however no actual ratios are provided. Please include these in the manuscript.

In section 3.2, the NK cell biology is quite interesting for discussion, however there is expression of multiple genes/cell surface markers but their relevance is not explained, such as DNAM-1. Rationale needs to be provided to guide the reader through the relevance of these molecules in the review.

Section 4.1 - Viral infections - HHV-6 - reactivation - please clarify "high incidence of HHV-6" and include % of patients affected. Furthermore, EBV and CMV infections should also be addressed as they also appear at increased frequency following UCBT.

Section 4.4 Age - it would be helpful to re-arrange this section to start with T cell reconstitution in children first, then adults.

Section 4.5 Cell dose: please specify exact cell dose ranges for sUBCT and dUCBT that correlate with engraftment and immune reconstitution in children and adults.

Section 5.1 Infection: please define the thresholds used to define CD4 and CD8 T cell reconstitution and also specify if the studies were performed in pediatrics or adults.

Section 5.2 Please clarify which malignancies suffer from increased risks of relapse post-UCBT - e.g. ALL vs. AML. In addition, the use of CBT for patients with MRD - Milano et al NEJM 2016 should be referenced in this section.

Author Response

This is an interesting and important review focused on the clinical relevance of immunobiology in UCBT. The review points out key differences in the immune cells in UCB and how the unique immune reconstitution (IR) following UCBT influences outcomes transplantation. The manuscript is nicely organized from discussing the composition of UCB, IR and outcomes following UCBT with specific focus on clinical factors that influence IR.

It is essential to highlight differences in the composition of the UCB graft compared to other transplant sources, such as PBSC (and specifically G-CSF-mobilized PBSC) and BM grafts, however the comparison is made to adult PB (not specified or mentioned re: G-CSF stimulation), so the comparison is not valid and should be revised for each lymphocyte subset appearing in section 2. In addition, as children and adults have different lymphocyte subsets, these should also be mentioned in the sections, to clarify whether the reference is to children vs. adult PBSCs vs. BM.

Response: We agree with the reviewer’s point that providing comparison in cell composition of grafts between UCB and other graft sources. Although several published data exist on comparison of immune reconstitution between UCB and other graft sources as described in section 3, there is paucity of data in the literature that provide direct comparison of NK or T cell composition of graft between UCB and other graft sources. As the purpose of section 2 is to provide overall ideas on immune cell composition of UCB, we believe that description of immune cell composition of UCB graft by comparing with normal adults’ PBMCs will meet the purpose well.

Figure 1 highlights the comparison, in which case it should include the composition of UCB, PBSC and BM. Re: the T cell subsets - naive vs. memory - it is not clear whether the pie charts refer to CD4 cells only or CD8 cells - it would be best to include both and also include other established T cells subsets, including stem cell memory, effector memory and terminal effector T cells.

Response: The chart on naïve vs. memory T cells concerns the entire T cell population as indicated in the figure 1. T cells subsets categorized by different axes than CD4 vs CD8, memory vs naïve T cells are described in the text. It is concerning that generating a figure on several subdivided subsets may confuse the readers. As stated above, the purpose of section 2 is to provide the conception on immune cell composition of UCB, not to highlight comparison with other sources of grafts. Hence, concise description of T and NK cell subsets in UCB graft in figure 1 will serve the purpose.

A table listing the immunophenotypes of the different cell types would be helpful to include in the review.

Response: Thank you for the comment. We believe that figure 2 may satisfy the reviewer’s comment.

The review mentions ratios of NK cells, and later, CD4 and CD8 T cells, however no actual ratios are provided. Please include these in the manuscript.

Response: We agree with the reviewer’s comment. The actual % from the literature is provided in the manuscript in section 2, and section 2.1.

In section 3.2, the NK cell biology is quite interesting for discussion, however there is expression of multiple genes/cell surface markers but their relevance is not explained, such as DNAM-1. Rationale needs to be provided to guide the reader through the relevance of these molecules in the review.

Response: We agree with the reviewer’s comment. We added further explanation on the relevance of NK cell phenotype in NK cell activity in section 3.2.

Section 4.1 - Viral infections - HHV-6 - reactivation - please clarify "high incidence of HHV-6" and include % of patients affected. Furthermore, EBV and CMV infections should also be addressed as they also appear at increased frequency following UCBT.

Response: We indicated the incidence of HHV-6 infection as 70-80% as reported in the cited literature. As this concise review is focusing on immune reconstitution and clinical outcomes, and HHV-6 infection is one of the most important factors on delayed immune reconstitution in UCBT, summarizing the relevance of HHV-6 will meet the purpose of our review. Moreover, CMV infection is discussed in section 5.1.

Section 4.4 Age - it would be helpful to re-arrange this section to start with T cell reconstitution in children first, then adults.

Response: Thank you for the comment. We tried to contrast adult T cell immune reconstitution (IR) with children T cell IR in UCBT. We are concerned that arranging this section by age will make the contrast more difficult.  

Section 4.5 Cell dose: please specify exact cell dose ranges for sUBCT and dUCBT that correlate with engraftment and immune reconstitution in children and adults.

Response: The focus of this review is on IR of UCBT, not engraftment.

Section 5.1 Infection: please define the thresholds used to define CD4 and CD8 T cell reconstitution and also specify if the studies were performed in pediatrics or adults.

Response: We indicated sufficient CD34+ cells to reach the “threshold” of thymopoiesis. We provided the threshold and study population in the first sentence of section 4.5. in the revised manuscript.  

Section 5.2 Please clarify which malignancies suffer from increased risks of relapse post-UCBT - e.g. ALL vs. AML. In addition, the use of CBT for patients with MRD - Milano et al NEJM 2016 should be referenced in this section.

Response: The disease-specific response is already stated in the manuscript. For example, “Successful CD4+ T cell reconstitution at day 100 after UCBT by less exposure to ATG is associated with lower NRM, lower relapse related mortality (particularly for AML), better event-free survival and better OS.68” and “Again, lower RI (HR=0.31, p=0.041) in association with improved T cell reconstitution was observed in AML.88” Unfortunately, Milano et al paper is cited in different sections, but not fit in this section as no immunological data such as IR is presented in the paper. Section 5.2. focuses on clinical outcomes in relation to immune reconstitution.

Reviewer 2 Report

Yun et al. reviewed the subject of immune constitution of the umbilical cord blood (UCB) graft and the immune reconstitution after UCB transplantation. Their review referenced 93 published work in the field and has done a thorough job summarizing the current knowledge about this subjects.

Major concerns: None

Minor concerns:

1) Figure 1 shows the cell composition of the UCB graft related to the text. While it was described in the text about its difference to other graft sources (PB), a visual representation of PB cell composition will be even better.

2) In Figure 3, I am not quite sure what the two large arrows on the far right represent? Effect on what?

3) Line 117-123, not sure why changed of font? Probably a format issue.

Author Response

Yun et al. reviewed the subject of immune constitution of the umbilical cord blood (UCB) graft and the immune reconstitution after UCB transplantation. Their review referenced 93 published work in the field and has done a thorough job summarizing the current knowledge about this subjects.

Major concerns: None

Minor concerns:

Figure 1 shows the cell composition of the UCB graft related to the text. While it was described in the text about its difference to other graft sources (PB), a visual representation of PB cell composition will be even better.

Response: We agree with the comment. We revised figure 1 as advised.

In Figure 3, I am not quite sure what the two large arrows on the far right represent? Effect on what?

Response: We agree with the reviewer’s comment. We revised Figure 3 changing “Net Effect” to “Net Effect on Immune Reconstitution”

Line 117-123, not sure why changed of font? Probably a format issue.

Response: Thank you for the comment. We changed the font and format.